# Melatonin Targets Metabolism in Head and Neck Cancer Cells by Regulating Mitochondrial Structure and Function

**DOI:** 10.3390/antiox10040603

**Published:** 2021-04-14

**Authors:** Ana Guerra-Librero, Beatriz I. Fernandez-Gil, Javier Florido, Laura Martinez-Ruiz, César Rodríguez-Santana, Ying-Qiang Shen, José M. García-Verdugo, Alba López-Rodríguez, Iryna Rusanova, Alfredo Quiñones-Hinojosa, Darío Acuña-Castroviejo, Jordi Marruecos, Tomás De Haro, Germaine Escames

**Affiliations:** 1Biomedical Research Center, Health Sciences Technology Park, University of Granada, 18016 Granada, Spain; aguerit@ugr.es (A.G.-L.); beatrizirenefg@correo.ugr.es (B.I.F.-G.); javiflorido@ugr.es (J.F.); Lauramartinezr@ugr.es (L.M.-R.); cesarsantana@ugr.es (C.R.-S.); shen@scu.edu.cn (Y.-Q.S.); alralba@correo.ugr.es (A.L.-R.); irusanova@ugr.es (I.R.); dacuna@ugr.es (D.A.-C.); 2CIBERFES, Ibs.Granada, San Cecilio University Hospital, 18016 Granada, Spain; 3Department of Physiology, Faculty of Medicine, University of Granada, 18016 Granada, Spain; 4Cavanilles Institute of Biodiversity and Evolutionary Biology, University of Valencia, 46980 Valencia, Spain; j.manuel.garcia@uv.es; 5Department of Biochemistry and Molecular Biology I, Faculty of Science, University of Granada, 18071 Granada, Spain; 6Department of Neurologic Surgery, Mayo Clinic, Jacksonville 32224, FL, USA; Quinones-Hinojosa.Alfredo@mayo.edu; 7Institut Català d’Oncologia (ICO), Hospital Universitari Dr. Josep Trueta, 17007 Girona, Spain; jmarruecos@iconcologia.net; 8UGC de Laboratorios Clínicos, Hospital Universitario San Cecilio, 18016 Granada, Spain; tomas.haro.sspa@juntadeandalucia.es

**Keywords:** melatonin, head and neck cancer cells, mitochondria, OXPHOS, glycolysis, mitophagy, apoptosis, free radicals

## Abstract

Metabolic reprogramming, which is characteristic of cancer cells that rapidly adapt to the hypoxic microenvironment and is crucial for tumor growth and metastasis, is recognized as one of the major mechanisms underlying therapeutic resistance. Mitochondria, which are directly involved in metabolic reprogramming, are used to design novel mitochondria-targeted anticancer agents. Despite being targeted by melatonin, the functional role of mitochondria in melatonin’s oncostatic activity remains unclear. In this study, we aim to investigate the role of melatonin in mitochondrial metabolism and its functional consequences in head and neck cancer. We analyzed the effects of melatonin on head and neck squamous cell carcinoma (HNSCC) cell lines (Cal-27 and SCC-9), which were treated with 100, 500, and 1500 µM of melatonin for 1, 3, and 5 days, and found a connection between a change of metabolism following melatonin treatment and its effects on mitochondria. Our results demonstrate that melatonin induces a shift to an aerobic mitochondrial metabolism that is associated with changes in mitochondrial morphology, function, fusion, and fission in HNSCC. We found that melatonin increases oxidative phosphorylation (OXPHOS) and inhibits glycolysis in HNSCC, resulting in increased ROS production, apoptosis, and mitophagy, and decreased cell proliferation. Our findings highlight new molecular pathways involved in melatonin’s oncostatic activity, suggesting that it could act as an adjuvant agent in a potential therapy for cancer patients. We also found that high doses of melatonin, such as those used in this study for its cytotoxic impact on HNSCC cells, might lead to additional effects through melatonin receptors.

## 1. Introduction

Head and neck squamous cell carcinoma (HNSCC), one of the most aggressive tumors, accounts for 80–90% of all malignant neoplasms in the oral cavity [1]. Despite advances in the treatments available, there is a low overall 5-year survival due to high recurrence. New anticancer therapeutic strategies are required to prevent or reverse the development of chemoresistance, which is a major obstacle to HNSCC management. Recent studies highlighted the important role played by tumoral metabolism in HNSCC [2,3,4]. Increasing evidence indicates that cancer could be considered a mitochondrial metabolic disease in which metabolic reprogramming plays a key role with regard to tumorigenesis and metastasis [5]. The energy needs of tumor cells are met through aerobic glycolysis, thus, limiting the transformation of glucose into acetyl-coenzyme A (acetyl-CoA). This in turn generates lactate and lowers the pH in the local environment, which play a key role in tumor cell migration and metastasis [6]. However, the maintenance of glycolysis when oxygen is present requires metabolic reprogramming which involves both epigenetic and genetic changes (the so-called Warburg effect) [7]. These changes, which are associated with a wide spectrum of metabolic adjustments, increases cell proliferation and self-renewal, as well as poor cell differentiation. Given the differences between cancer and normal cell metabolism, it is vital to precisely elucidate the molecular mechanisms involved in mitochondrial metabolism reprogramming and to provide effective drug repositioning therapeutic strategies to target metabolic reprogramming.

Many essential mitochondrial functions play a much more important role in cancer metabolism than previously thought [8]. Cancer cell mitochondria are novel therapeutic targets given their structural and functional differences from their normal cell counterparts. Increased production of reactive oxygen species (ROS) also generates mutations involving malignant transformation and cancer progression, thus making these organelles promising targets of anticancer therapy. However, ATP production in normal cells might be disrupted by the targeting of mitochondrial ATP production, while mitochondrial ROS-mediated signaling in immune cells might be disrupted when mitochondrial redox status is targeted [9]. To resolve this problem, cancer cell mitochondria need to be targeted without damaging normal cell mitochondria [5].

Melatonin (N-acetyl-5-methoxytryptamine) is a potent anti-inflammatory and antioxidant molecule. Moreover, melatonin attracts attention as an oncostatic agent. However, the precise mechanism of the action involved in the oncostatic anti-neoplastic impact of melatonin is not fully elucidated. The involvement of mitochondria in the anti-proliferative effect of melatonin on neoplastic cells is of particular relevance for the development of innovative cancer treatments [10,11,12,13,14]. Indeed, we found that melatonin amplifies the cytotoxic effects of chemotherapy and radiotherapy in HNSCC, which are associated with changes in mitochondrial respiratory capacity and ROS balance in these cells [10,11]. Changes in mitochondrial respiration are well-known to contribute to metabolic remodeling, although the mechanism of action involved in melatonin-driven tumoral death remains unclear. Moreover, previous studies showed that melatonin not only enhances the oncostatic effects of radio- or chemotherapy on tumor cells, but also protects normal cells against the adverse effects of these treatments [10,12,14,15].

In this study, we tested the hypothesis that the effects of melatonin on cancer cells can be explained by bioenergetic stress, resulting from the inability of tumor cells to switch on glycolysis, in order to produce ATP, following the inhibition of oxidative phosphorylation (OXPHOS). To investigate this hypothesis, we targeted tumor cell metabolism in HNSCC following melatonin treatment. We confirmed the cytotoxic effect of melatonin on HNSCC and its involvement in a change of metabolism in these cancer cells and also determined the precise mechanism of action involved. We aimed to understand the mechanisms that lead to tumor progression in HNSCC and to provide evidence for melatonin treatment strategies for HNSCC that is aimed at the targeting metabolism.

## 2. Materials and Methods

### 2.1. Cell Culture and Treatment

Head and neck squamous carcinoma cells Cal-27 (ATCC: CRL-2095) and SCC-9 (ATCC: CRL-1629) were obtained from the Cell Bank of the Scientific Instrumentation Centre of the University of Granada. Cal-27 cells were grown, as previously described [10], in Dulbecco’s Modified Eagles Medium-GlutaMAX (DMEM, high glucose, GlutaMAX™ Supplement, Fisher Scientific, 41965039, Madrid, Spain) with 10% fetal bovine serum (FBS) (16000044; Fisher Scientific Madrid, Spain), 1 mM sodium pyruvate, and 2% antibiotic/antimycotic solution (15240062, Fisher Scientific, Madrid, Spain). SCC-9 cells were cultured in DMEM-F12 Nutrient Mixture Ham medium (11320033; Fisher Scientific, Madrid, Spain) supplemented with 10% FBS, 2% antibiotic/antimycotic solution, 2 mM L-glutamine (25030081; Fisher Scientific, Madrid, Spain), 0.5 mM sodium pyruvate, and 0.4 μg/mL hydrocortisone. Both cell lines were maintained in a humidified 37 °C incubator with 5% CO_2_. Cal-27 and SCC-9 cells were negative for mycoplasma, tested using a PCR detection kit, following the manufacturer’s instructions (25235; LiliF Diagnostics, Burlington, MA, USA).

Melatonin (33457-24; Fagron Ibérica S.A.U., Terrasa, Spain) stock solution was prepared in 15% propylene glycol (PG; 24414.296; VWR, Radnor, PA, USA) in phosphate buffer solution (PBS) (14190-094, Life Technologies) and filter-sterilized through a 0.2 μm pore filter (Sartorius Biotech GmbH, Gottingen, Germany). Cells were grown to 60–70% confluence and serum starved for 24 h. They were then treated with different melatonin concentrations (0, 100, 500, and 1500 μM) for 24, 72, and 120 h (1, 3, and 5 days). Vehicle was added to the control group.

### 2.2. Metabolomic Analysis

The TCA cycle metabolites, citric acid, succinyl-CoA, acetyl-CoA, NADH, and lactic acid, were quantitatively determined by ultraperformance liquid chromatography, in combination with triple quadrupole mass spectrometry (UPLC-MS/MS), at the Center for Omic Sciences (Rovira i Virgili University, Tarragona, Spain). Since metabolic changes are observed in a short period of time, cells were treated with melatonin only for 6 h for metabolomic analysis. After harvesting the cells, the metabolites were extracted by a liquid–liquid extraction with hexane. Samples were analyzed in a 1290 Infinity LC Series chromatograph coupled with a 6495 iFunnel Triple Quad MS/MS (Agilent Technologies, Santa Clara, USA) equipped with an AQUITY UPLC HSS T3 1.8 μm, 2.1 × 100 mm column (Waters, Milford, USA). The mobile phase used was 20 mM NH4AC and 0.2% acetic acid as solvent A and acetonitrile (ACN) as solvent B. The column flow was 0.5 mL/under the following mobile phase gradient conditions—0–1 min 0% B isocratic, 5 min 50% B, 6–9 min 100% B isocratic, and 10 min 0% B with a post run of 3 min. The volume of injection was 5 μL. Ionization was performed using a source of electrospray (ESI), with a temperature and flow of Drying Gas (N_2_) of 290 °C and 18 L/min respectively, a nebulization gas (N_2_) pressure of 20 psi and a sheath gas (N_2_) temperature and flow of 350 °C and 10 L/min. The fragment or voltage was 380 V, capillary voltage was 3500 V in the positive and 3000 in the negative and the nozzle voltage was 750 V in the positive and 1500 V in the negative mode. Detection was performed in the QqQ detector through acquisition of Multiple Reaction Monitoring (MRM). Quantification of samples was performed through interpolation of the obtained area in the standard curve with the matrix match method.

### 2.3. Pyruvate Measurement

Intracellular pyruvate concentration was measured using a colorimetric assay kit (EPYR-100, BioAssay Systems, Hayward, CA, USA) in a microplate reader spectrophotometer (Power Wave X-1; Bio-Tek Instruments, Inc., Winooski, VT, USA) at 570 nm, according to the manufacturer’s instructions Data were normalized by the protein content measured by Bradford assay.

### 2.4. Mitochondrial Respiration

The oxygen consumption rate (OCR) was determined using the Seahorse Extracellular Flux XF-24 analyzer (Seahorse Bioscience, N.Billerica, MA, USA). Cells were seeded in XF 24-well cell culture microplates (Seahorse Bioscience) at a density of 8 × 10^4^ cells/well and incubated overnight at 37 °C with 5% CO_2_. Prior to performing the assay, the medium was changed to Base Medium (102353-100, Seahorse Bioscience, N. Billerica, MA, USA) containing 10 mM glucose, 5 mM pyruvate, and 2 mM glutamine, and the cells were equilibrated for 1 h at 37 °C without CO_2_.

The OCR was measured as previously described [16], both at the basal level and in response to modulators of maximal mitochondrial electron transport system (ETS) capacity. A total of 1 µM oligomycin and 1 µM trifluorocarbonylcyanide phenylhydrazone (FCCP) (0.5 µM in two injections) were injected, followed by 1 µM rotenone/antimycin A. Inhibition of ATP synthase by oligomycin was used to determine the proportion of OCR used for ATP production under basal conditions. Maximal uncoupled respiration was assessed after the addition of FCCP, and the residual nonmitochondrial OCR was measured using rotenone/antimycin A. The average of four readings per cell line was recorded, and it was normalized by the number of cells. Cells were counted in a hemocytometer, and viability was determined by trypan blue staining.

### 2.5. Measurement of Redox Status

ROS production was measured using 100 µM of 2′ 7′ dichlorofluorescin diacetate (DCFH-DA) probe (Sigma-Aldrich, Madrid, Spain), according to the protocol described by Shen et al. [10]. DCFH-DA was transformed into the fluorescent DCF inside the cell. Fluorescence was then measured with the microplate fluorescence reader FLx800 (Bio-Tek Instruments, Inc., Winooski, VT, USA) for 30 min every 5 min, at 485 nm to excitation and 530 nm to emission wavelengths [10]. Superoxide dismutase (SOD) activity was assayed in terms of its ability to inhibit the auto-oxidation of adrenalin to adrenochrome at pH 10.2. Oxidation of adrenalin was measured at 490 nm for 10 min at 30 °C as Misra et al. described [17]. SOD activity was expressed as U/mg prot (1 unit = 50% inhibition of epinephrine auto-oxidation).

### 2.6. Glycolysis Capacity

The extracellular acidification rates (ECAR) were measured using the Seahorse Extracellular Flux XF-24 analyzer (Seahorse Bioscience, N. Billerica, MA, USA). Cells were plated in XF24 cell culture microplates at a density of 8 × 10^4^ cells/well in DMEM. After 24 h, the culture medium was changed to Base Medium (102353-100, Seahorse Bioscience, N. Billerica, MA, USA) supplemented with 2 mM glucose and was incubated at 37 °C for 1 h in a CO_2_-free incubator. The extracellular acidification rate (ECAR) was measured using the XFp Glycolysis Stress Test Kit (Seahorse Bioscience), according to the manufacturer’s instruction. Accordingly, after baseline measurement, the following injections were made—10 mM glucose, 1 μM oligomycin, and 50 mM 2-DG. ECAR values were normalized to the cell number. Cells were counted in a hemocytometer and viability was determined by trypan blue staining.

### 2.7. Electron Microscopy Analysis

Electron microscopy analysis was carried out according to the protocol described by Cebrian et al. [18]. Cells were fixed in 3% glutaraldehyde in PB buffer 0.1 M pH 7.4 and postfixed in 2% OsO_4_ and stained in 2% uranyl acetate in the dark. Finally, cells were dehydrated in ethanol, rinsed with propylene oxide (Lab Baker, Deventry, Holland) and embedded overnight in Araldite (Durcupan, Sigma). Upon polymerization, embedded cultures were detached from the chamber slide and glued (Super glue, Loctite) to Araldite blocks. Semi-thin sections (1.5 μm) were cut with an ultramicrotome (Ultracut UC-6, Leica, Heidelberg, Germany) mounted onto slides and stained with 1% toluidine blue. Ultrathin sections (50 nm) were obtained with an ultramicrotome using a diamond tipped knife (Ultra 45° Diatome) and stained with lead citrate (Reynolds Solution).

Images were obtained with a transmission electron microscope (FEI Tecnai Spirit BioTwin, Hillsboro, OR, USA) using a digital camera (Morada, Soft Imaging System, Olympus, Tokyo, Japan). At least 10 independent cells were studied for each condition. Among them, 20 mitochondria were analyzed for each cell culture, belonging up to 2 mitochondria to the same cell. The length of the mitochondrial crests per mitochondrial area was quantified using the ImageJ software (La Jolla, California, USA).

### 2.8. Quantification of Mitochondrial Mass

Mitochondrial mass was related to the cardiolipin content. Quantification of cardiolipin was performed using the fluorescent probe Nonyl Acridine Orange (NAO, A1372; Fisher Scientific, Madrid, Spain), according to the protocol described by Shen et al. [10]. The fluorescence was read using a microplate fluorescence reader FLx800 (Bio-Tek Instruments, Inc., Winooski, VT, USA), at an excitation wavelength of 485 nm and emission of 530 nm.

### 2.9. Mitochondrial DNA Copy Number

Human mitochondrial DNA was quantified by real-time PCR in a Stratagene Mx3005P qPCR System (Agilent Technologies, Inc., Santa Clara, CA, USA), as described by Lopez LC et al. [19]. We used primers and probe complementary to sequences of the nuclear gene 18 S (TaqMan Gene Expression Assays Hs99999901_s1, Applied Biosystems) and to the sequences of the mtDNA 12 S gene. mtDNA values were normalized to nDNA data (mtDNA/nDNA ratio).

### 2.10. Cell Proliferation Assay

Cell viability was measured using an MTT assay (MTT, V13154; LifeTechnologies, Madrid, Spain), based on the conversion of soluble MTT (yellow) to non-soluble formazan (purple). Cells were seeded in a 96-well culture plate (167008, Thermo Scientific, Madrid, Spain) and MTT assay was performed following the manufacturer’s instructions.

### 2.11. Cell Cycle Analysis

Cellular DNA content was measured by flow cytometry using a propidium iodide (PI)/RNase kit (PI/RNASE, Immunostep, Salamanca, Spain) staining, according to the manufacturer’s instructions. Data were analyzed using flow cytometry in a Becton Dickinson FACSCanto II cytometer (Madrid, Spain). Data were reported as the percentage of cells in each phase of the cell cycle.

### 2.12. Western Blot Analysis

Protein extraction and Western blot analyses were performed as described previously [20], and Bradford assay was used to determine protein concentration. Adequate amounts of proteins were separated on 7% and 12.5% PhastGel homogeneous gels (GE Healthcare Life Sciences, Barcelona, Spain). Proteins were transferred to Hybond™-ECL nitrocellulose membranes (GE Healthcare Life Sciences, Barcelona, Spain) using the PhastSystem (GE Healthcare Life Sciences, Barcelona, Spain). Next, the membranes were first incubated in blocking buffer containing 5% BSA in PBS plus 0.1% Tween 20 and were then, incubated with the primary antibodies diluted in blocking buffer overnight at 4 °C. The utilized primary antibodies included OXPHOS Rodent WB Antibody Cocktail (1:1000; ab110413; Abcam, Inc., Cambridge, MA,USA), OPA1 (1:200; sc393296; Santa Cruz Biotechnology, Heidelberg, Germany), MFN2 (1:200; sc5033; Santa Cruz Biotechnology), Drp1 (1:200; sc-32898; Santa Cruz Biotechnology), LETM (1:200; sc-134672; Santa Cruz Biotechnology), Bax (1:200; sc-526; Santa Cruz Biotechnology), Bcl2 (1:200; sc-492; Santa Cruz Biotechnology), NIX (1:1000; N0399; Sigma-Aldrich, Madrid, Spain), LC3 (1:1000; nb100-2220; Novus, Abigdon, UK), ATG12 (1:200; sc-271688; Santa Cruz Biotechnology), MT1 (1:500; TA321736; OriGene Technologies, Rockville, MD, USA), MT2 (1:500; TA314217; OriGene Technologies), RORα (1:500; TA311450; OriGene Technologies), and GAPDH (1:500; sc-166574; Santa Cruz Biotechnology). Anti-mouse at dilution 1:1000 (554002; BD Biosciences Pharmigen, San Jose, CA, USA), anti-rabbit at dilution 1:5000 (31460; Thermo Scientific, Madrid, Spain), and anti-goat diluted 1:1000 (sc-2768; Santa Cruz Biotechnology) HRP conjugated secondary antibodies were used according to the manufacturer’s instructions.

The images were analyzed using the ECLTM Prime Western Blotting Detection Reagent (GE Healthcare Life Sciences, Barcelona, Spain), with the Kodak Image Station 2000R (Eastman Kodak Company, Rochester, NY, USA), and quantified using 1D Image Analysis software 3.6. Protein band intensities were normalized to GAPDH and data were expressed as the percentage relative to controls.

### 2.13. Statistical Analysis

Data are expressed as the mean ± S.E.M of a minimum of three independent experiments. Statistical analysis was performed with the Prism software (GraphPad, La Jolla, CA, USA), using one-way analysis of variance (ANOVA) with Bonferroni test. Normality test was performed by a Shapiro-Wilk analysis, showing that all date had a normal distribution. A *p*-value of < 0.05 was considered to indicate statistical significance.

## 3. Results

### 3.1. Melatonin Up-Regulates Key TCA Cycle Metabolites in HNSCC Cells

The role of melatonin in regulating cancer cell metabolism remains largely unexplored. To further dissect the effects of melatonin in cancer metabolism, we carried out metabolomic analyses using UPLC-MS/MS, which found clear differences between the control group and the melatonin-treated cells (Figure 1). We found that after 6 h, melatonin-treated cells upregulated key TCA cycle metabolites, including acetyl CoA, citric acid, and succinyl-CoA (Figure 1A–C), which led to increased NADH levels (Figure 1D) and reduced levels of pyruvate (Figure 1E). A larger influx of pyruvate into the mitochondria plus the increased NADH levels would lead to enhanced oxidative phosphorylation. These two processes would lead to mitochondrial ROS production. Interestingly, the intracellular lactate increased, compared to the control (Figure 1F).

These findings corroborated the important role played by melatonin to induce a change of head and neck cancer cell metabolism. To independently validate these data, we then assessed the mitochondrial respiration and glycolytic capacity.

### 3.2. Melatonin Treatment Induces Uncoupling between Respiration and Phosphorylation in Mitochondria, Correlating with Increased ROS Production in HNSCC Cells

To further determine whether the inhibition of metabolic reprogramming in HNSCC cells after melatonin treatment was linked to their mitochondria activity, functional mitochondrial analyses were carried out using the Seahorse XF24 extracellular flux analyzer in Cal-27 cells (Figure 2A–C) and in SCC-9 cells (Figure 2D–F). Consistent with previous findings [10,11], we observed elevated OCR corresponding to increased basal respiration (Figure 2G–I) and maximal respiratory capacity of the electron transport system (ETS), with melatonin treatment of Cal-27 cells in a dose- and time-dependent direct manner (Figure 2J–L). However, at high 500 and 1500 µM doses of melatonin during 3 and 5 days of treatment, melatonin reduced the OCR (Figure 2H,I,K,L), suggesting that the mitochondrial function was defective during long treatments with higher concentrations of melatonin. SCC-9 cells showed no significant differences in basal respiration and ETS capacity (Figure 2G,H), but exhibited a significant decrease in mitochondrial respiration, after 3 and 5 days of melatonin treatment at a dose of 1500 µM (Figure 2H,I,K,L). Thus, confirming the presence of defective mitochondria at high doses of melatonin. No significant changes in ATP turnover were observed in either group tested, following melatonin treatment (Figure 2M–O).

We then examined OXPHOS protein expression levels by Western blotting (Figure 3). In Cal-27 cells, protein analysis showed that treatment with melatonin led to a significant increase in the expression of complexes I, II, III, and IV, relative to the control cells, especially after 1 and 3 days of treatment (Figure 3A–D). The rate of increase in their expression levels fell after 5 days of treatment (Figure 3E,F), which was consistent with the results for mitochondrial respiratory function. However, no differences in the expression levels of complex V or ATP synthase were observed in any melatonin-treated groups (Figure 3). This correlated with mitochondrial ATP turnover, which remained unchanged (Figure 2M–O). Although SCC-9 cells were more resistant than Cal-27 cells, its values indicated a partial uncoupling between respiration and phosphorylation in mitochondria, which could correlate with an increase in ROS production. Thus, we found that melatonin treatment was able to induce significant levels of ROS production in Cal-27 cells, in a dose- and time-dependent manner (Figure 4A,B). However, ROS production decreased after 5 days of melatonin treatment (Figure 4C), which correlated with the decrease in mitochondrial function. The increase in ROS levels was an indication of the weak scavenging ability of cancer cells. We therefore measured superoxide dismutase (SOD) activity (Figure 4D–F), which constituted the first line of defense against excess ROS. SOD activity was found to significantly decrease following treatment with melatonin, in a dose- and time-dependent manner, thus, confirming that melatonin increased oxidative stress in cancer cells.

### 3.3. Effects of Melatonin on Glycolytic Activity in HNSCC Cells

Cancer cells are able to use different energy sources. In order to clarify the metabolic flux of HNSCC treated with melatonin, we focused on glycolytic activity, due to its prime role in cancer cell proliferation. Our investigation of the effects of melatonin on glycolytic activity in HNSCC cells showed a decrease in the extracellular acidification rate (ECAR), after 3 and 5 days of treatment with high doses of melatonin (Figure 5A–F). Cellular glycolytic reserve capacity, calculated by subtracting the maximal ECAR from the basal rate, increased significantly, following melatonin treatment, reaching a peak at a dose of 500 µM, after 1 and 3 days of treatment, as compared to the basal control group (Figure 5G–I).

With regard to the effect of melatonin treatment on HNSCC glycolysis, hexokinase II (HK-II) protein levels were determined. Treatment with 500 µM of melatonin significantly decreased the hexokinase II (HK-II) protein content in Cal-27 cancer cells, after 5 days of treatment (Figure 5J–L).

### 3.4. Melatonin Treatment Modifies the Mitochondrial Morphology of HNSCC Cells

To determine whether the changes in mitochondrial function observed in HNSCC cells correlated with compromised mitochondrial structural integrity, we evaluated mitochondrial morphology by electron microscopy (EM). As shown in Figure 6A, mitochondria size was larger in Cal-27 cells treated with melatonin, as compared to the control group. Similar results were observed for the SCC-9 cells (images not shown). Mitochondria in the control group showed a more spheroid and ovoid morphology with poor cristae, as compared to the elongated mitochondria with well-defined cristae in melatonin-treated cells, especially at higher melatonin concentrations of 500 and 1500 µM, indicative of a high metabolic rate. Interestingly, the effects of melatonin on Cal-27 cells, which became evident after one day of treatment, were maintained throughout the treatment (Figure 6A–D). However, melatonin had a more marked impact on SCC-9 cells than on Cal-27 cells, only after three days of treatment, with no effect being observed after five days (Figure 6C,D). These results were consistent with the increase in mitochondrial mass, as compared to the control group, especially at a concentration of 1500 µM, after one day of treatment in Cal-27 cells (Appendix A) and at concentrations of 500 and 1500 µM after three days of treatment in both cell lines (Appendix A). We again observed that long-term treatment with high doses of melatonin damaged mitochondria (Appendix A).

In addition, it was reported that during reprogramming there was not only a reduction of mitochondrial mass, but also in mitochondrial DNA (mtDNA) [21]. Consistent with the comparable increase in mitochondrial mass, we observed a sharp increase in mtDNA (Appendix A), especially after one day of treatment with 1500 µM of melatonin in Cal-27 cells (Appendix A). However, three days of treatment with high concentrations of melatonin led to a progressive increase in the mtDNA copy number, while the results for mtDNA in SCC-9 cells were in line with those for the mitochondrial mass (Appendix A).

### 3.5. Melatonin Alters Mitochondrial Fission and Fusion in HNSCC Cells

To explore the mechanism underlying melatonin-induced mitochondrial elongation in tumor cells, we analyzed the expression levels of the core components in mitochondrial fission/fusion machinery by Western blot. We measured the inner mitochondrial membrane protein, optic atrophy1 (OPA1), and outer mitochondrial membrane protein, mitofusin 2 (MFN2), which control mitochondrial fusion, the dynamin-related protein 1 (Drp1), is mainly involved in mitochondrial fission [22], and the leucine zipper/EF hand-containing transmembrane-1 (LETM1), which acts as an anchor protein for complex formation between mitochondria and ribosome, and as a regulator of mitochondrial biogenesis (Figure 7) [23].

Melatonin decreased OPA1 and MFN2 levels in Cal-27 cells after three and five days of treatment, as compared to the control group (Figure 7A–F,M), with a peak effect observed at a concentration of 500 µM. In addition, Drp1 protein levels increased at concentrations of 500 and 1500 µM after one day of treatment, and decreased after 5 days of treatment (Figure 7G–I,M). Similar results were observed with regard to LETM1 (Figure 7J–M) [24].

Overall, these findings indicate that melatonin increases mitochondrial fission after 24 h, which then decreases after 5 days of treatment. While melatonin decreased fusion, the downregulation of mitochondrial fission was reported to increase cancer cell apoptosis [25]. On the other hand, the percentage of apoptotic cells increased as OPA1 and MFN2 gene expression levels fall [26].

### 3.6. Melatonin Increases Apoptosis and Autophagy in HNSCC Cells

We evaluated the expression levels of apoptotic proteins by Western blot. Melatonin increased Bax (Figure 8A–C,J) and decreased Bcl-2 (Figure 8D–F,J) expression levels, with an increase also observed in the Bax/Bcl-2 ratio (Figure 8G–I), following dose- and time-dependent melatonin treatment, reaching a peak effect after three and five days at concentrations of 1500 or 500 µM, respectively, in Cal-27 cells.

In addition, oxidative stress is well-known to promote autophagy, which boosts the mitophagic removal of damaged mitochondria [27]. We then analyzed autophagy/mitophagy activation and determined ATG12-ATG5, LC3-II, and NIX protein levels (Figure 8K–T).

Interestingly, ATG12-ATG5 protein levels gradually decreased after three days of melatonin treatment, at concentrations of 500 and 1500 µM (Figure 8R,T). In contrast, LC3-II protein levels increased during all dose-dependent melatonin treatments (Figure 8N–P,T), which also induced an accumulation of NIX proteins (Figure 8K–M,T).

Thus, melatonin increased NIX- and LC3-dependent mitophagy, while ATG-dependent autophagy remained unchanged.

### 3.7. Melatonin Decreases HNSCC Cell Proliferation and Induces Cell Cycle Arrest in the G1 and G2/M Phases

As mitochondria are essential for cancer cell proliferation, with several cell signaling mitochondria known to be involved in controlling cell cycle progression [28], we measured cell proliferation by the MTT assay. We observed morphological changes such as cell detachment and shrinkage, especially at high concentrations of melatonin (500 and 1500 µM) after 3 and 5 days of treatment (Figure 9A); cellular proliferation was suppressed by dose- and time-dependent treatment with melatonin as compared to the control group (Figure 9B–D), with maximal effect achieved at a concentration of 1500 µM, although SCC-9 cells were found to be more resistant to treatment than Cal-27 cells.

To determine whether the inhibition of proliferation induced by melatonin is involved in the cell cycle arrest, we evaluated its impact on the Cal-27 cell cycle by flow cytometry. As shown in Figure 9E–H, concentration- and time-dependent treatment with melatonin significantly increased the percentage of cells in the G1 and G2/M phases and decreased Cal-27 cell proliferation in phase S of the cell cycle.

These results showed that melatonin had selective anti-proliferative effects on HNSCC cells.

### 3.8. High Doses of Melatonin Change Melatonin Receptor Protein Levels

Given that the functions of melatonin are mediated by both receptor-dependent and receptor-independent mechanisms [29], we then explored the involvement of melatonin receptors in the cytotoxic effects of melatonin on HNSCC by Western blot analysis (Appendix A).

Incubation of cells with melatonin significantly increased MT1 gene expression after one and three days of dose-dependent treatment (Appendix A). However, MT1 expression decreased after five days of treatment with 500 µM of melatonin (Appendix A). These data are consistent with the cellular damage produced by long treatments with high doses of melatonin.

On the other hand, dose- and time-dependent melatonin treatment led to a marked decrease in MT2 and RORα levels (Appendix A–J).

## 4. Discussion

Metabolic reprogramming by cancer cells to rapidly adapt to stress conditions such as hypoxia and limited nutrients is now avowed as hallmarks of cancer, with several recent studies shedding light on the role of mitochondria in this process [5,30]. The integration of tumor cell metabolism reprogramming into novel cancer therapies attracts considerable interest in the cancer research community [31,32,33,34,35], with mitochondrial metabolism, in particular, being an important potential target for therapeutic intervention [5,36].

On the whole, cancer cells use glycolysis rather than mitochondrial respiration, even in presence of sufficient oxygen, to produce ATP molecules [37]. Our findings indicate that glycolysis inhibition and increased OXPHOS activity could explain, at least partially, the inhibition of metabolic reprogramming by melatonin in HNSCC cells. OXPHOS deficiency also closely correlates with the invasive and metastatic properties of tumor cells [38]. Our results conclude that melatonin treatment increases TCA intermediates such as acetyl-CoA, citric acid, and succinyl-CoA, as well as NADH, but decreases pyruvate levels. The influx of pyruvate into mitochondria, which connects glycolysis, with no oxygen requirement, to mitochondrial oxidative phosphorylation, is a critical metabolic process. Pyruvate directly drives the metabolic flux of the TCA cycle forward, through conversion to acetyl-CoA by pyruvate dehydrogenase, with the condensation of acetyl-CoA with oxaloacetate leading to the formation of citrate. In addition to increasing acetyl-CoA and citrate levels, melatonin could counteract reductions in mitochondrial pyruvate carrier (MPC) tumor cell activity and in pyruvate oxidation rates [39]. Pyruvate is also shunted away from the cancer cell mitochondrial metabolism through its reduction to lactate. Lactate dehydrogenase A (LDHA) reduces pyruvate to lactate and, at the same time, oxidizes NADH to NAD^+^, thus explaining the low levels of cancer cell NADH. Melatonin increased both NADH and intracellular lactate levels, suggesting that extracellular lactate transport is inhibited. The transport of lactate out of the cell acidifies the tumor microenvironment, allowing progression to metastasis due to suppression of local immune response and activation of metalloproteases [8,40]. Melatonin was found to increase intracellular lactate, suggesting that lactate transport is inhibited following melatonin treatment.

We decided to determine the impact of melatonin on the behavior of mitochondria in cellular metabolism. Our results showed that melatonin stimulates oxygen consumption, basal respiration, and ETS capacity, and increases OXPHOS protein expression levels. Surprisingly, mitochondrial intracellular ATP turnover did not increase in either Cal-27 or SCC9 cell lines following melatonin treatment. Complex V (ATP synthase) protein expression was unaffected despite an overall increase in OXPHOS markers, such as complex I-IV and respiratory function. These results could be attributed to the uncoupling of mitochondrial electron transport activity and phosphorylation. Both cellular respiration and ATP production were selectively affected in tumor cells, indicating that the mitochondrial bioenergetic inhibition was a selective mechanism caused by melatonin. In addition, incomplete oxidative phosphorylation in the electronic transport chain results in an increase in ROS. Melatonin was observed to increase ROS production in a dose- and time-dependent manner, which was partly due to decreased mitochondrial SOD activity. High levels of mitochondrial SOD are well-known to greatly stimulate glycolysis through H_2_O_2_ production and activation of specific signaling pathways and to strongly inhibit mitochondrial oxidative metabolism [41]. However, the complex role of ROS in mediating tumor progression remains a subject of controversy. Several studies concluded that ROS, as signaling molecules, facilitate cancer stem cell (CSC) self-renewal and expansion [42], stimulated cell invasiveness [43], and potentiated tumor progression. Nonetheless, increased ROS levels were also reported to be detrimental to other cancer cells and to inhibit cancer metastasis [42]. Furthermore, restoration of OXPHOS activity and increased levels of ROS can sensitize CSCs to chemotherapeutic drugs [44].

Glycolysis is generally used by cancer cells for ATP production rather than mitochondrial respiration, even in the presence of adequate oxygen [37]. In addition, resistant cells can acquire a glycolytic metabolism to survive cytotoxic stress [38]. Given that glycolytic marker expression in HNSCC is associated with poor survival rates and resistance to chemo and radiotherapy [2], we observed that melatonin decreases glycolysis. On the whole, these findings indicate that glycolysis inhibition and increased OXPHOS might explain, at least partly, the inhibition of metabolic reprogramming induced by melatonin in HNSCC cells [38].

Mitochondrial metabolic reprogramming is shown to be necessary for tumorigenesis, metastasis, and stemness [5,30]. Guido et al. suggested that remodeling of mitochondrial morphology protects tumor cells against cellular stress with remarkable success, and therefore, that adjustment of mitochondrial morphology could be a potential tool in human cancer treatment [45]. It was suggested that mitochondrial dynamic changes act as sensors of energy stress and initiators of cancer cell metabolic reprogramming [46]. However, the effects of melatonin on reprogrammed metabolism mediated by the changes in cancer cell mitochondrial morphology are little understood. Our study showed that melatonin induced mitochondrial elongation and facilitated the formation of cristae in HNSCC cells and thus increased mitochondrial mass. The shape of cristae was also demonstrated to regulate the stability and assembly of respiratory chain supercomplexes, which affected respiratory efficiency [27]. It was possible to speculate that mitochondrial elongation correlated with a clear metabolic shift from glycolysis to oxidative phosphorylation.

Mitochondrial functions were closely associated with the morphodynamics of mitochondria, consisting of both fusion and fission processes. In general, fission, whose pro- or anti-apoptotic function remains a matter of debate, is thought to be pro-tumorigenic [25]. On the other hand, some authors showed that an increase in activation of mitochondrial fusion in cancer is a metabolic advantage to maintain tumor growth [26]. In this study we found that melatonin decreased fusion and fission after 5 days of treatment. Our results appear to be congruent with the finding that mitochondrial fusion proteins are responsible for maintaining mitochondrial membrane structures, matrix homogeneity, and mitochondrial genome integrity, which are vital for cell survival [47]. In addition, the loss of fusion/fission balance is linked to mitochondrial and cellular dysfunctions such as apoptosis. Other studies demonstrated that certain anti-tumor drugs activate autophagy while inducing apoptosis [10,48]. This is in line with our finding that melatonin triggers apoptosis in a dose- and time-dependent manner. Melatonin leads to a decrease in anti-apoptotic factors and to a release of pro-apoptotic factors from the mitochondrial inter-membrane space into the cytosol. This could be partly due to severe impairment of the mitochondrial electron transport chain, which is a major site of ROS production [49].

ROS production triggers mitochondrial autophagy and protein degradation in order to mitigate the cellular stress induced by melatonin. However, autophagy, which plays a dual role, can induce senescence or cell death as a tumor suppressor and can also induce tumorigenesis [50]. Reprogrammed cancer cells, undergoing glycolysis, display mitochondrial dysfunction and activate autophagy/mitophagy [51]. Although the specific mechanism of autophagy involved in cancer cells remains largely unknown, Atg5-dependent autophagy was shown to be transiently activated early on the reprogramming process [52], which was blocked in Agt5^−/−^ cells in particular. This was consistent with our results which showed that melatonin decreased glycolysis and ATG5 levels after three days of treatment, although melatonin was also found to induce mitophagy and apoptosis.

We found that the cytotoxic effects of melatonin in decreasing cell proliferation are dose- and time-dependent. We show that high doses of melatonin, such as those used in this study and elsewhere [20], with their cytotoxic effect on HNSCC cells, further increased melatonin receptor expression, which might be involved, at least partly, in the indoleamine effect [12].

## 5. Conclusions

In conclusion, we showed that melatonin increased mitochondrial TCA intermediates and OXPHOS levels, resulting in an inhibition of metabolic reprogramming of aerobic glycolysis. Overall, the study provides new insights into the mechanisms underlying melatonin’s anti-tumor activity in head and neck cancer cells. To the best of our knowledge, this was the first study to demonstrate that the anti-proliferative effects of melatonin in HNSCC are, at least partially, due to an inhibition of metabolic remodeling, resulting in enhanced mitochondrial function. This, in turn, leads to increased oxidative stress and activation of apoptosis and mitophagy in HNSCC cells.

## Figures and Tables

**Figure 1 antioxidants-10-00603-f001:**
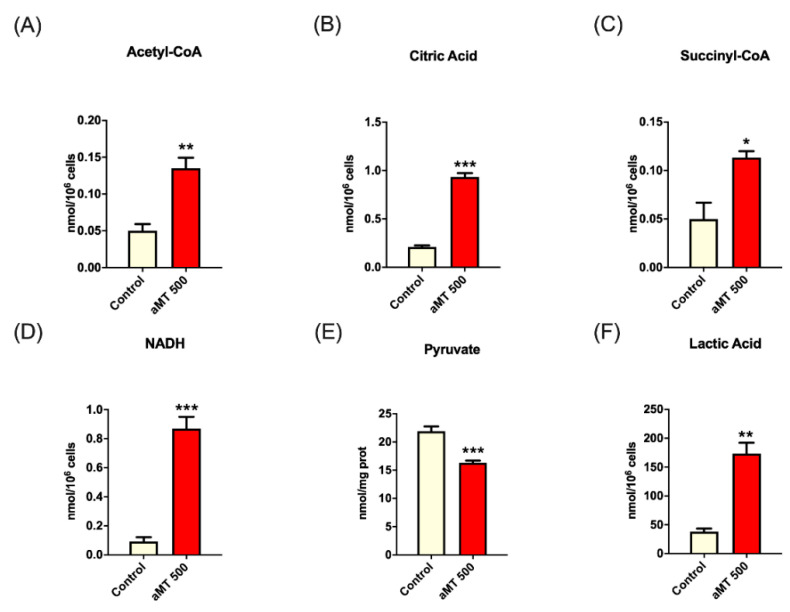
Upregulation by melatonin of key TCA cycle metabolites in HNSCC cell line Cal-27. Metabolomic study of intracellular levels of acetyl-CoA (**A**), citric acid (**B**), succinyl-CoA (**C**), NADH (**D**), and lactate (**F**). Pyruvate levels were measured using a colorimetric test (**E**). Treatment groups included vehicle (control) and melatonin (aMT) at a concentration of 500 µM. *n* = 4 per group. Data are presented as mean ± SEM. * *p* < 0.05, ** *p* < 0.01, *** *p* < 0.001 vs. control group.

**Figure 2 antioxidants-10-00603-f002:**
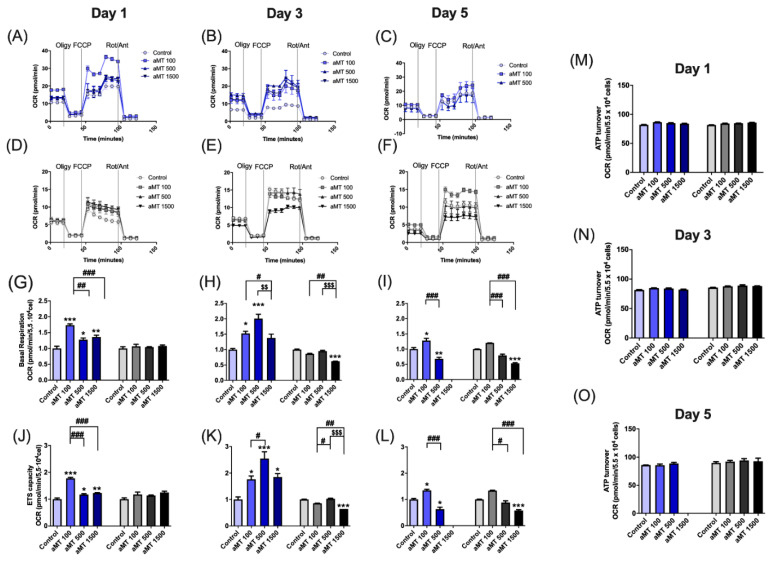
Effect of melatonin on mitochondrial respiration in HNSCC cell lines Cal-27 (in blue) and SCC-9 (in black and grey). Oxygen consumption rate (OCR) after 1 day (**A**,**D**), 3 days (**B**,**E**), and 5 days (**C**,**F**) of melatonin treatment, basal respiration (**G**–**I**), maximal respiratory capacity (ETS) (**J**–**L**) and ATP turnover (**M**–**O**). Treatment groups include vehicle (control) and melatonin (aMT) at concentrations of 100 µM, 500 µM, and 1500 µM. Data for aMT 1500 group are not shown at day 5 because most cells died. *n* = 6 per group. Data are presented as mean ± SEM. * *p* < 0.05, ** *p* < 0.01, *** *p* < 0.001 vs. control; # *p* < 0.05, ## *p* < 0.01, ### *p* < 0.001 vs. aMT 100 µM group; $$ *p* < 0.01, $$$ *p* < 0.001 vs. aMT 500 µM group.

**Figure 3 antioxidants-10-00603-f003:**
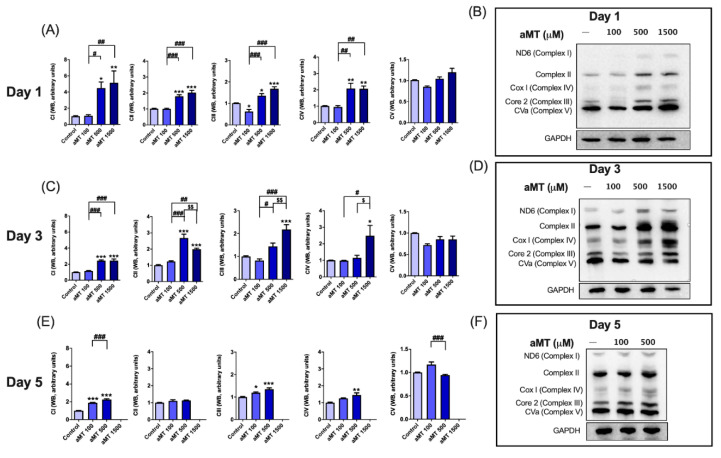
Regulation of OXPHOS protein expression by melatonin in Cal-27 cells. Analysis of OXPHOS protein expression by Western blot after 1 day (**A**,**B**), 3 days (**C**,**D**), and 5 days of melatonin treatment (**E**,**F**). Data for aMT 1500 group are not shown at day 5 because most cells died. *n* = 6 per group. Data are presented as mean ± SEM. * *p* < 0.05, ** *p* < 0.01, *** *p* < 0.001 vs. control; # *p* < 0.05, ## *p* < 0.01, ### *p* < 0.001 vs. aMT 100 µM group; $ *p* < 0.05, $$ *p* < 0.01 vs. aMT 500 µM group.

**Figure 4 antioxidants-10-00603-f004:**
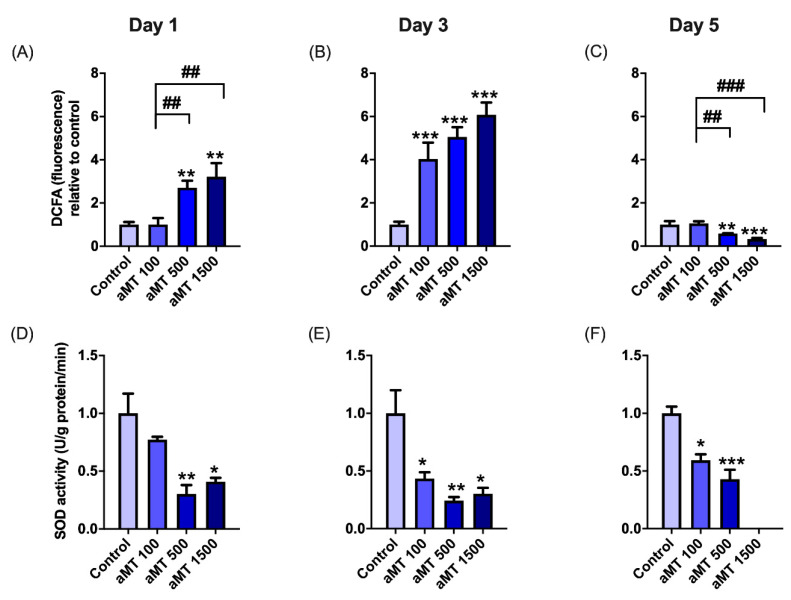
Melatonin-induced oxidative stress in Cal-27 cells. Measurements of intracellular ROS levels by fluorometry after staining with the DCF fluorescent probe (**A**–**C**) and SOD activity (**D**–**F**). Data for aMT 1500 group are not shown at day 5 because most cells died. *n* = 6 per group. Data are presented as mean ± SEM. * *p* < 0.05, ** *p* < 0.01, *** *p* < 0.001 vs. control; ## *p* < 0.01, ### *p* < 0.001 vs. aMT 100 µM group.

**Figure 5 antioxidants-10-00603-f005:**
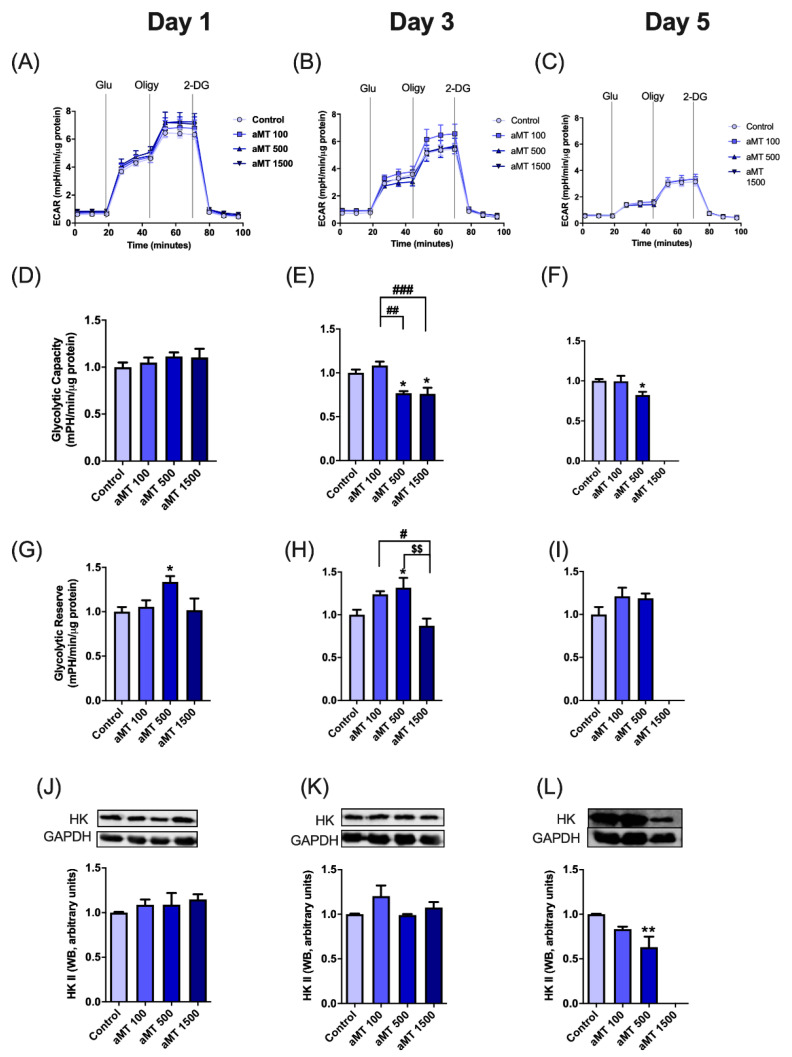
Effect of melatonin on glycolysis in Cal-27 cells. Extracellular acidification rate (ECAR) (**A**–**C**), glycolytic capacity (**D**–**F**), glycolytic reserve (**G**–**I**) analyzed by Seahorse, and hexokinase II protein expression analyzed by Western blot (**J**–**L**). Data for the aMT 1500 group are not shown at day 5 because most cells died; *n* = 6 per group. Data are presented as mean ± SEM. * *p* < 0.05, ** *p* < 0.01 vs. control; # *p* < 0.05, ## *p* < 0.01, ### *p* < 0.001 vs. aMT 100 µM group; $$ *p* < 0.01 vs. aMT 500 µM group.

**Figure 6 antioxidants-10-00603-f006:**
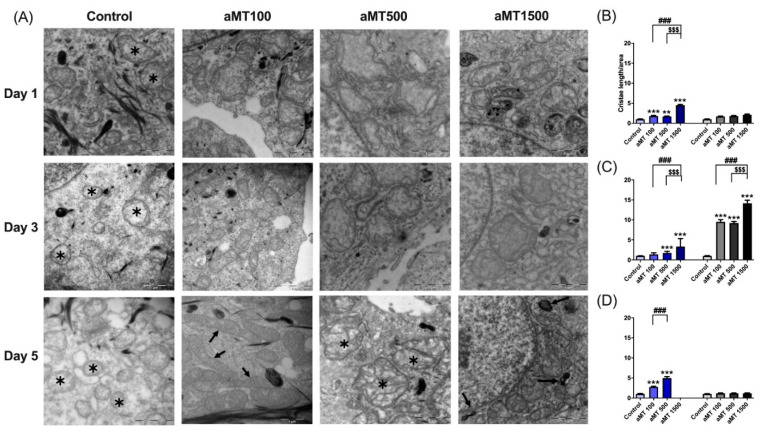
Melatonin-induced changes in mitochondrial morphology. Alterations in mitochondrial morphology analyzed by EM in Cal-27 cells (**A**), cristae length analyzed using the Image J software (**B**–**D**) in Cal-27 (in blue) and SCC-9 cells (in black and grey). In the control cells, the typical morphology of the Cal-27 mitochondria could be observed, with a round shape and few mitochondrial cristae (asterisks). No changes were detected over time. With aMT 100 concentration, mitochondria tend to get smaller and adopt elongated shapes, especially after 5 days of treatment (arrows). At dose of aMT 500, mitochondria do not show much change between day 1 and day 3, but at 5 days, mitochondria can be seen with a marked increase in cristae number (asterisks). At a concentration of 1500, at one day of treatment, mitochondria are similar to those in the control cells, although heterogeneous dense bodies are occasionally appreciated (arrows). After 3 days of treatment, the mitochondria maintain their shape and appearance, with abundant cristae. After 5 days, dense bodies continue to be seen (arrows) and it is difficult to detect cristae in some mitochondria. *n* = 6 per group. Data are presented as mean ± SEM. ** *p* < 0.01, *** *p* < 0.001 vs. control; ### *p* < 0.001 vs. aMT 100 µM group; $$$ *p* < 0.001 vs. aMT 500 µM group. Scale bar = 1 µm.

**Figure 7 antioxidants-10-00603-f007:**
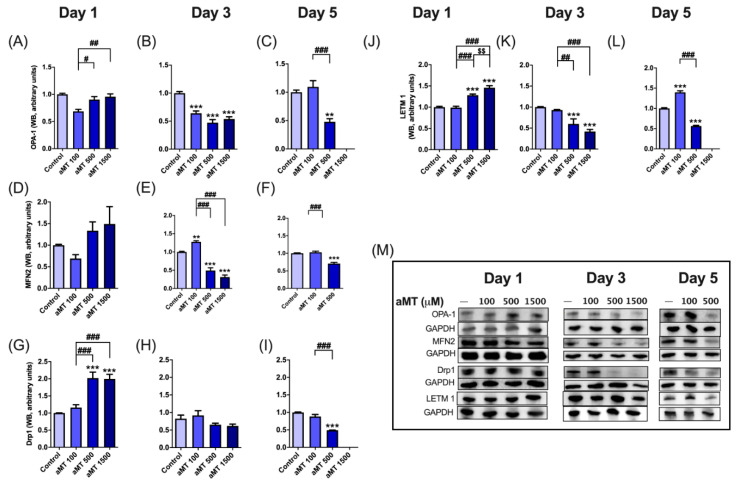
Melatonin-altered mitochondrial dynamics in Cal-27 cells. Western blot analysis of OPA-1 (**A**–**C**,**M**), MFN2 (**D**–**F**,**M**), Drp1 (**G**–**I**,**M**), and LETM1 (**J**–**L**,**M**). Data for aMT 1500 group are not shown at day 5 because most cells died. *n* = 6 per group. Data are presented as mean ± SEM. ** *p* < 0.01, *** *p* < 0.001 vs. control; # *p* < 0.05, ## *p* < 0.01, ### *p* < 0.001 vs. aMT 100 µM group; $$ *p* < 0.01 vs. aMT 500 µM group.

**Figure 8 antioxidants-10-00603-f008:**
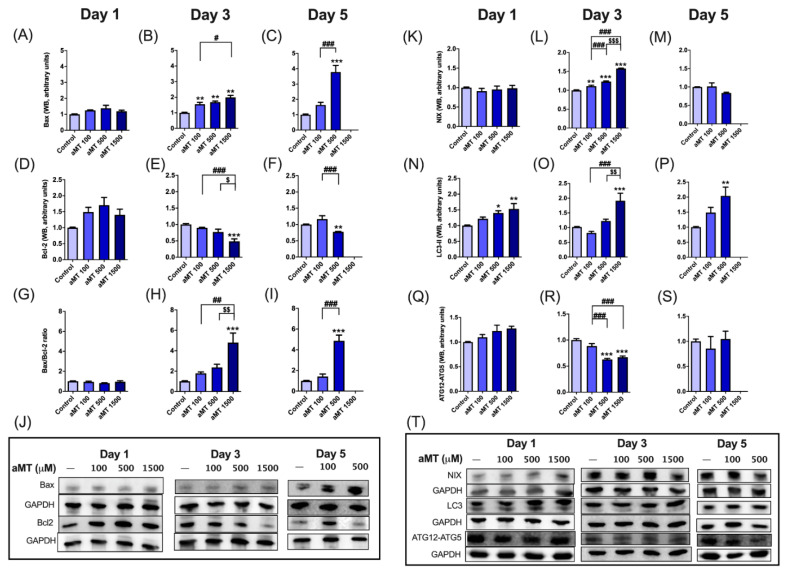
Increases in apoptosis and autophagy caused by melatonin in Cal-27 cells. Western blot analysis of Bax (**A**–**C**,**J**), Bcl-2 (**D**–**F**,**J**), Bax/Bcl-2 ratio (**G**–**I**), NIX (**K**–**M**,**T**), LC3 (**N**–**P**,**T**), and ATG12-ATG5 (**Q**–**T**). Data for aMT 1500 group are not shown at day 5 because most cells died. *n* = 6 per group. Data are presented as mean ± SEM. * *p* < 0.05, ** *p* < 0.01, *** *p* < 0.001 vs. control; # *p* < 0.05, ## *p* <0.01, ### *p* < 0.001 vs. aMT 100 µM group; $ *p* < 0.05, $$ *p* < 0.01, $$$ *p* < 0.001 vs. aMT 500 µM group.

**Figure 9 antioxidants-10-00603-f009:**
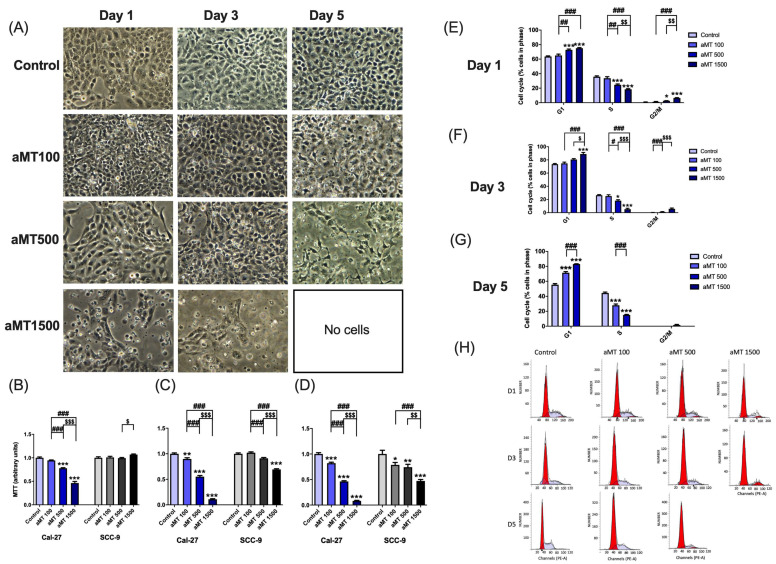
Decreased proliferation of Cal-27 and SCC-9 cells caused by melatonin. Morphological alterations (**A**), proliferation rate (**B**–**D**), percentage of cells in each cell cycle phase (**E**–**G**), and representative plots showing cell redistribution (**H**). Data for aMT 1500 group are not shown at day 5 because most cells died. *n* = 6 per group. Data are presented as mean ± SEM. * *p* < 0.05, ** *p* < 0.01, *** *p* < 0.001 vs. control; ## *p* < 0.01, ### *p* < 0.001 vs. aMT 100 µM group; $ *p* < 0.05, $$ *p* < 0.01, $$$ *p* < 0.001 vs. aMT 500 µM group.

## Data Availability

Data is contained within the article.

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
