# Peer review of "Melatonin Targets Metabolism in Head and Neck Cancer Cells by Regulating Mitochondrial Structure and Function"

_antioxidants, 2021, doi:10.3390/antiox10040603_

Round 1

Reviewer 1 Report

In their article “Melatonin reverses metabolic reprogramming in head and neck cancer cells by regulating mitochondrial structure and function” Guerra-Librero and colleagues provide a novel potential explanation for the anti-proliferative effects of melatonin on head and neck squamous cell carcinoma through the alteration of cell metabolism. 

The paper is well written and easy to follow. However, some questions need to be addressed before the manuscript is accepted for publication.

  • Introduction:
    • The authors need to expand the section about the relevance of using melatonin for cancer treatment to introduce the reader to the topic. Currently in Line76, the authors state that the precise mechanism of action involcevd in the oncostatic anti-neoplastic impact of melatonin has not been fully elucidated, and not much more is said about the background about melatonin. 
  • Methods:
    • Line 150, describe the method used for protein content analysis.
    • Line 163, describe how the number of cells was determined. Same applies to line 182.
    • Line 227, describe what was used to determine “adequate amounts of proteins”.
  • Results:
    • Figure 1. Please explain why a colorimetric test was needed for the detection of pyruvate. Wasn’t the metabolite detected by UPLC-MS/MS? 
    • Figure 1. The authors conclude that the results support a role for melatonin in the inhibition of cancer metabolism reprogramming. Since no normal control cells were used as a baseline in the experiments (only untreated or melatonin-treated cancer cells), it is difficult to argue that an inhibition of reprogramming has occurred. What is certain is that melatonin upregulates key metabolites in the TCA cycle. Authors need to rephrase “inhibition of cancer metabolism reprogramming”.
    • Figure 2. The figure calling and figure legend need to be clearer, making it easy for the reader to distinguish between the two cell lines in the different panels.
    • Figure 2. Can the authors provide an explanation as to why the basal respiration of the control group for one of the lines decreases by 50% between day 1 and days 3 and 5, and almost doubles at day 3 for the other cell line? Same question for the ETS. Perhaps consider presenting normalized results towards the control group.
    • Figure 2. In the text the authors claim a dose- and time-dependent regulation of the OCR and ETS. Please specify if you mean a direct or reverse correlation.
    • Figure 2. Panels M-O, did the authors measure ATP or are they plotting the differences between groups after injection of oligomycin? If the latter, please revise lines 308-309 in the main text.
    • Figure 3. Data for the highest dose is missing at day 5. Can the authors provide an explanation? Did the cells die? If the same data is available for the second line, SCC-9, consider including it in the supplement.
    • Figure 4. Why is data missing for the highest dose in panel F? Same comment as before, please include data for SCC-9 in the supplement if available.
    • Figure 5. To study the impact of melatonin treatment on glycolysis, the authors measured the ECAR in one cell line. However, without direct measurement of lactate it is rather impossible to tell to what extent glycolysis is the main responsible for the total acidification rate. Include the data for SCC-9 in the supplement.
    • Figure 6. Specify in the figure legend what cell line is shown in panel A. Label the data for the different cell lines properly in panels B-D. The reader should not have to assume that the first 4 bars belong to Cal-27 and the rest to SCC-9.
    • Figure 7. Include the data for SCC-9 in the supplement.
    • Figure 8. Revise lines 427-428, NIX- and LC3 dependent what?

In general, since the results are based on only two cancer lines, the authors are encouraged to include the data from both cell lines if it is available.

  • Discussion:
    • Lines 495-500, lactate was never directly measured by the authors. Please rephrase.
    • Same goes for lines 503-505, ATP was never directly measured either.
    • Line 564 “the autophagic mechanism of autophagy”, perhaps substituted with “the specific mechanism of autophagy”.

Throughout the manuscript the authors keep alluding to an inhibition of metabolic reprogramming. As mentioned above, it is difficult to tell if indeed there is an inhibition of metabolic reprogramming or if metabolic reprogramming by melatonin is occurring since no control non-cancer lines are used as a defined baseline. In general the authors need to rephrase inhibition of cancer metabolic reprogramming throughout the manuscript, what the data actually displays is a change of metabolism in head and neck cancer cells in response to melatonin treatment.

Reviewer 2 Report

The topic of the presented study is of first importance. Melatonin has been found to have oncostatic effect, but mechanisms of action are still under investigation. During last years, the impact of melatonin on the mitochondrial function has been considered. Constantly increased morbidity and mortality of cancer, including neck and head cancers, worldwide makes every research in this area very interesting. The manuscript is written in good scientific English, the study was planned correctly, methodology was described in details, the results are presented in clear way with proper figures, schemes and photos. The discussion is very interesting and corresponds with other studies. I recommend to publish the manuscript with no changes.

Author Response

Thank you very much for your positive comments.